Biological and growth parameters of Plotosus lineatus in the Mediterranean Sea

http://orcid.org/0000-0003-2939-5838 Doğdu Servet Ahmet 1 servetdogdu@yandex.com
Turan Cemal 2
1 Underwater Technology, Iskenderun Technical University , Hatay, Iskenderun , Türkiye
2 Molecular Ecology and Fisheries Genetics Laboratory, Marine Science Department, Faculty of Marine Science and Technology, Iskenderun Technical University , Hatay, Iskenderun , Türkiye
Yapıcı Sercan
Electronic publication date: 2024 Feb 21
Publication date: 2024
Volume: 12
Electronic Location ID: e16945
Received 2023 Nov 21; Accepted 2024 Jan 24
Copyright: © 2024 Doğdu and Turan
Copyright year: 2024
Copyright holder: Doğdu and Turan
License: This is an open access article distributed under the terms of the Creative Commons Attribution License, which permits unrestricted use, distribution, reproduction and adaptation in any medium and for any purpose provided that it is properly attributed. For attribution, the original author(s), title, publication source (PeerJ) and either DOI or URL of the article must be cited.
License URL: https://creativecommons.org/licenses/by/4.0/

Keywords: Plotosus lineatus, Age-lenght, Growth parameters, Eel catfish, Invasive species, Mediterranean Sea

Funding: The authors received no funding for this work.

==============================
This study examined the age distribution and growth characteristics of the striped eel catfish (Plotosus lineatus), which is an invasive alien species in the eastern Mediterranean. A total of 1,011 samples were collected from Iskenderun Bay (Turkey), with lengths ranging from 5.1 to 16.8 cm, predominantly comprising females (1:1.92). Age 3 represented the majority in the population (52.03%). The value of the scaling exponent “b” of the length-weight relationship was less than “3” for both sexes (females: 2.28; males: 2.26; combined: 2.27). The results for the von Bertalanffy growth parameters were observed for the combined sexes as, L∞ = 24.9934 cm, k = 0.1718 year−1, and t₀ = −1.7707 years. The striped eel catfish populations in Iskenderun Bay exhibited negative allometric growth patterns and were predominantly composed of adult individuals. This study presents the dataset on the length-weight correlations, age-growth characteristics, and von Bertalanffy growth parameters of Plotosus lineatus in the Mediterranean Sea, thereby significantly contributing to comprehending the stock dynamics. It is anticipated that this study will make a significant contribution to the management of P. lineatus stocks, given its invasive nature.

Introduction

The opening of the Suez Canal and the effects of climate change have led to a rise in the population of alien and invasive species in the Mediterranean Sea (Turan, Ergüden & Gürlek, 2016; Tiralongo et al., 2020; Turan & Doğdu, 2022). To date, more than 100 Indo-Pacific fish species have been entered into the eastern Mediterranean basin via the Suez Canal (Galil, Marchini & Occhipinti-Ambrogi, 2018; Turan et al., 2018; Azzurro & D’Amen, 2022; Mutlu et al., 2023).

The family Plotosidae represented of single species in the Mediterranean Sea by striped eel catfish Plotosus lineatus (Thunberg, 1787), which are 10 accepted genera and 42 accepted species all over the world (Fricke, Eschmeyer & Van der Laan, 2023). P. lineatus has an extensive distribution that includes the Indo-Pacific region, the Red Sea and the Mediterranean Sea. Striped eel catfish was first documented in the Mediterranean from the coast of Israel by Golani (2002) and has rapidly established itself as a prominent component of the benthic fauna in the Mediterranean Sea. Subsequently, the second recorded occurrence was reported off the Egyptian coast near El-Arish, the third recorded sighting was documented from the Syrian coast near Tartous, the fourth record was recorded in Turkish marine waters and the fifth record was reported in Northern Cyprus (Temraz & Ben Souissi, 2013; Ali, Saad & Soliman, 2015; Doğdu et al., 2016; Tiralongo et al., 2022). It was observed that the species showed a significant population increase within a few years after its first appearance in the Mediterranean Sea. In a monitoring study conducted by Turan & Doğdu (2023), it was reported that P. lineatus rapidly increased in number after it was first seen in 2016 and became the leading species along the eastern coast of the Mediterranean Sea in Türkiye.

The determination of age and growth for a species is essential as it provides critical input data for the assessment of marine fish stocks and helps in understanding their biology (Hilborn, 1992; Vieira et al., 2009; Tiralongo et al., 2018; Ergüden & Doğdu, 2020). Until today, there is only one publication available on the age-growth development and population dynamics of the striped eel catfish (Vijayakumaran, 1998). No research has been carried out on the biological parameters (age-growth) of P. lineatus in the Mediterranean Sea. The publications related to the striped eel catfish are primarily migration reports (Temraz & Ben Souissi, 2013; Ali, Saad & Soliman, 2015; Doğdu et al., 2016; Tiralongo et al., 2022; Turan & Doğdu, 2022; Bayhan & Ergüden, 2022). On the other hand, recent studies deal with its growth performance study (Asriyana et al., 2021), length–frequency (Mehanna, 2023), length-weight relationships (Edelist et al., 2012; Ueng et al., 2022) and clinical cases (Bentur et al., 2018; Turan et al., 2020).

The primary aim of this study is to the first information regarding the age, growth, and certain biological parameters of the invasive alien fish species, the striped eel catfish (Plotosus lineatus), which has established itself in Iskenderun Bay in the eastern Mediterranean.

Materials and Methods

Between January 2022 and July 2023, a total of 1,011 specimens of P. lineatus were collected in Iskenderun Bay, Turkey. All samples were collected by fishermen from a commercial trammel net at night. All samples were captured at depths ranging from 20 to 40 m. The collected samples were transported to the laboratory and stored on crushed ice for analysis. In the laboratory, each fish was assessed, with the total length (TL) measured in centimeters (cm), weight (W) recorded to the nearest 0.01 Gram, and the sex was determined through macroscopic observation of the gonads. Strictly abide by the ethical principles of experimental animal welfare, the Fisheries Animal Experiments Local Ethics Committee of the Iskenderun Technical University has fully approved this study (ISTE-06112023-115215).

Age identification was performed by analyzing the left and right sagittal otoliths. A stereo zoom microscope with 10× magnification was employed for the age determination process. The number of opaque zones in the otoliths was counted to determine the age of each fish, as shown in Fig. 1. These have been independently analysed three times by different readers for truthfulness.

Figure 1 Photograph of a sectioned sagittal otolith from a 162 mm, TL female individual of Plotosus lineatus a 4-year-old in Iskenderun Bay.

The length-weight relationship (L-W-R) was established following the formula of Ricker (1975):

W=aLb

The formulation abbreviations are as follows; W is weight (g), L is total length (cm), b is the length-weight factor and a is a constant. The L-W-R was measured for all samples, and separate analyses were conducted for males and females. A t-test was used to compare the slopes of the regression lines and the mean lengths for each sex (Zar, 1999). Statistical analysis of data was performed using SPSS 22.0.

The von Bertalanffy function was established following the formula of Von Bertalanffy (1938):

Lt=L∞(1−e−k(t−t0))

The formulation abbreviations are as follows: Lt represents the total length at time t, K is the growth constant L∞ is the asymptotic length, and t0 is the theoretical age at length 0.

The growth performance index was determined using the formula outlined by Pauly & Munro (1984), which serves as a length-based indicator of growth performance.

⊘=log10K+2log10L∞

The length at first maturity (TL50) was estimated, along with ±95% confidence intervals, by constructing a maturity curve. This curve was created by fitting data points within the middle-class interval and taking into account the percentage of mature fish corresponding to each length class interval. TL50 was subsequently calculated as the length at which a randomly chosen individual has a 50% probability of being mature, effectively representing the midpoint of both the x and y axes on the maturity curve (Somerton, 1980; Roa, 1999). Growth parameters were computed using RStudio Team (2020) in combination with the FLR (Fisheries Library in R) platform (Sparre, Ursin & Venema, 1989; Kell et al., 2007).

Results

A total of 1,011 samples, with total lengths ranging from 5.1 to 16.8 cm, were collected. The sagittal otoliths were examined, and the maximum age was determined to be 4 years for individuals of both sexes. Age 3 was the dominant age group in the population, accounting for 52.03% of the total. The mean total length and total weight were observed as 13.36 ± 1.73 cm and 15.53 ± 6.14 g, respectively (Table 1).

Table 1 Parameters of length-weight relationships of Plotosus lineatus.

Sex	N	Total length min–max (cm)	Weight
(g)	W = a × TLb	
a	b	R2	
Females	665	5.1–16.8	2.9–45.5	0.0739	2.2821	0.9944	
Male	346	5.7–16.7	4.1–45.3	0.0782	2.2598	0.9952	
Combined	1,011	5.1–16.8	2.9–45.5	0.0756	2.2737	0.9947	

The length-weight relationship of P. lineatus was calculated as W = 0.0739 × L2.2821 (R2 = 0.9944) for females, W = 0.0782 × L2.2598 (R2 = 0.9952) for males, and W = 0.0756 × L2.2737 (R2 = 0.9947) both combined sexes (Table 1 and Fig. 2) and significantly differed between the sexes (p < 0.001). The “b” value being less than “3” for all individuals indicates that this species exhibits negative allometric growth.

Figure 2 Length-weight relationships of Plotosus lineatus as (A) female, (B) male (C) combined.

The population was composed of 65.77% females (665 individuals) and 34.23% males (346 individuals). The majority of the population consisted of females (1:1.92). It was dominated by females. The χ2 test demonstrated a significant difference between the expected theoretical 1:1 ratio and the observed proportion (P < 0.001).

The age groups of P. lineatus ranged from 0 to 4 years for individuals of both sexes. Age 3 (52.03 %) was dominant in the population. The descriptive statistics and estimated parameters of the length-weight relationship are provided in Table 2, and the age frequency distribution of all individuals can be found in Fig. 3.

Table 2 Mean total length values for each age group of P. lineatus.

Age	Combined sexes	Females	Males	
N	%	TLmean ± SD (TLmin−TLmax) (cm)	N	%	TLmean ± SD (TLmin−TLmax) (cm)	N	%	TLmean ± SD (TLmin−TLmax) (cm)	
0	16	1.58	6.55 ± 1.03 (5.1–8.1)	8	1.20	6.46 ± 1.21 (5.1–8.1)	8	2.31	6.65 ± 0.88 (5.7–7.8)	
1	30	2.97	9.93 ± 0.59
(8.4–10.4)	19	2.86	9.72 ± 0.66
(8.4–10.4)	11	3.18	10.29 ± 0.10
(10.2–10.4)	
2	231	22.85	11.43 ± 0.40
(10.5–12)	149	22.40	11.44 ± 0.40
(10.5–12)	82	23.70	11.41 ± 0.42
(10.5–12)	
3	526	52.03	13.52 ± 0.93
(12.1–14.9)	363	54.59	13.56 ± 0.91
(12.1–14.9)	163	47.11	13.43 ± 0.98
(12.1–14.9)	
4	208	20.57	15.79 ± 0.46 (15.2–16.8)	126	18.95	15.81 ± 0.46 (15.2–16.8)	82	23.70	15.76 ± 0.46 (15.2–16.7)	
Total	1,011	100	13.26 ± 1.93 (6.55–16.8)	665	100	13.37 ± 1.710 (8.4–16.8)	346	100	13.35 ± 1.77 (10.2–1.67)	

Figure 3 Age-length distribution of P. lineatus.

The growth parameters, estimated using the von Bertalanffy equation, have been presented for females, males, and both combined sexes in Table 3 and Fig. 4. The von Bertalanffy growth parameter results were obtained for combined sex L∞ = 24.9934 mm; k = 0.1718 year−1; t0= −1.7707 year.

Table 3 von Bertalanffy growth parameters (L∞, k and t0), growth performance index (Ø) and length-weight relationships parameters (a, b and R2) for present and previous studies on Plotosus lineatus.

References	L∞ (cm)	k (/y)	t0	Ø	a	b	R2	
Vijayakumaran (1998)	C: 24.373	C: 1.3694	C: 0.0085	C: 2.9105	F: 0.00000079	F: 3.4295	F: 0.9930	
M: 0.0000021	M: 3.3478	M: 0.9951	
Edelist et al. (2012)	–	–	–	–	C: 0.0034	C: 3.266	C: 0.9655	
Ueng et al. (2022)	−	−	−	−	F: −5.3629	F: 3.0917	F: 0.8124	
M: −6.0395	M: 3.2914	M: 0.9097	
C: −5.8245	C: 3.2391	C: 0.9833	
Mehanna (2023)	F: 29.40	F: 0.48	−	C: 2.68	−	−	−	
M: 31.05	M: 0.44	
C: 31.82	C: 0.40	
This study	C: 24.9934	C: 0.1718	C: −1.7707	C: 2.3915	F: 0.0739	F: 2.2821	F: 0.9944	
M: 0.0782	M: 2.2598	M: 0.9952	
C: 0.0756	C: 2.2737	C: 0.9947	

Figure 4 von Bertalanffy growth curve for P. lineatus for all samples.

The logistic equation for the maturity ogive was fitted, resulting in an estimated TL50 (length at first maturity) of 13 cm for the combined sexes of P. lineatus in Iskenderun Bay. The logistic model indicated a rapid maturity process occurring between 12 and 14 cm TL (Fig. 5).

Figure 5 Maturity ogives estimation (TL50) of P. lineatus for both sexes with a 95% confidence interval (—).

Discussion

This study provides the first dataset on length-weight relationships, age-growth dynamics, and von Bertalanffy growth parameters, making significant contributions to the understanding of stock dynamics for the striped eel catfish, Plotosus lineatus, in Iskenderun Bay within the Mediterranean Sea.

The coefficient of determination (R2) for P. lineatus was observed as 0.9944 for females, 0.9952 for males and 0.9947 for combined sexes. The length-weight relationship (LWR) displayed a strong correlation between the length and weight of the specimens. Vijayakumaran (1998) studied the coefficient of determination (R2) of P. lineatus for females 0.9930 and for males 0.9951 in Visakhapatnam, India. Edelist et al. (2012) studied the coefficient of determination (R2) of P. lineatus for unsexes 0.9655 in the Mediterranean coasts of Israel. Ueng et al. (2022) found the coefficient of determination (R2) of P. lineatus was observed for females 0.8124, males 0.9097 and combined sexes 0.9833 in southern Taiwan. The present study showed the highest correlation with previous findings (Table 3). The differences in R² can be attributed to various factors, including gonad maturity, sex, diet, stomach fullness, health, and preservation techniques. These factors are known to influence length-weight relationships (LWRs) in fish, and it’s important to note that none of these factors were considered in this study (Karachle & Stergiou, 2008).

The “b” values obtained from the length-weight relationship equations were 2.28 for females, 2.25 for males, and 2.27 for both combined sexes. It is worth noting that these values are the lowest observed compared to previous studies conducted on P. lineatus (Table 3). Vijayakumaran (1998) described the “b” value for females as 3.42 and for males as 3.34. Edelist et al. (2012) observed the “b” value for P. lineatus for unsexes as 3.26. Ueng et al. (2022) found the b value for P. lineatus for females 3.09, for males 3.29 and combined sexes for 3.23. Our study showed negative allometric growth. However, previous studies showed positive allometric growth. The variations in “b” values can be attributed to a range of factors, including differences in environmental conditions, biological parameters, and sampling methods. These differences may encompass variations in size ranges, the number of individuals collected, and the sampling period. The length-weight ratio can indeed vary for the same species in different geographical areas, and this variability may be influenced by various environmental factors. These factors can include the quality, quantity, and size of the available food, spawning time, salinity, temperature, habitat, gonadal maturity, season, presence or absence of young individuals, sex, health of the fish, timing of fishing, and the type of fishing gear used (Ricker, 1975; Safran, 1992; Turan et al., 2021).

Growth parameters such as L∞ (asymptotic length), k (growth rate), and t0 (theoretical age at length 0) are essential statistics employed in various models to assess and analyze the current status of fish stocks. These parameters provide valuable insights into the growth and age structure of fish populations, which are crucial for fisheries management and conservation efforts. Also, these statistics are valuable for comparing fish growth not only between different species but also for assessing the growth of the same species at various times and in different geographic locations. They serve as essential tools for understanding the dynamics of fish populations and for making informed decisions in fisheries management and conservation (Maunder & Punt, 2013). The present study observed length (L∞), growth rate (k), and age at zero length (t0) of P. lineatus for combined sexes as 24.9 cm, 0.17 and −1.77, respectively. The current values of L∞, k, and t0 were compared with those obtained from previous studies of P. lineatus, as detailed in Table 3. The asymptotic length (L∞) observed in the present study was relatively similar to that of previous studies. This similarity may suggest that there are favourable environmental conditions for P. lineatus in the Mediterranean Sea, promoting consistent growth patterns for this species. It’s worth noting that the t0 parameter for both sexes of P. lineatus in the current study was notably lower compared to previous studies conducted in different locations. This difference in t0 may reflect variations in the early life history and growth patterns of the species in the specific environment of the study area (Table 3). The observed decrease in t0 values may be due to the absence of juvenile specimens. It should not be ignored that this species may be an economic contributor to Mediterranean fisheries as in other regions where it is distributed (Manikandarajan et al., 2014; Mehanna, 2023).

The maturity ogives (TL50) was calculated at 13 cm for combined sexes. Figure 4 shows a fast maturity process between 12 and 14 cm in total length. These results for TL50 recognize that age at maturity and first sexual maturity may vary by region, but it is important to adopt a minimum capture size of at least TL50 (13 cm) for P. lineatus economic recovery.

Conclusions

This study provides the first dataset on the length-weight relationship, age-growth characteristics, and von Bertalanffy growth parameters, making significant contributions to our understanding of the stock dynamics of the striped eel catfish, Plotosus lineatus, in Iskenderun Bay within the Mediterranean Sea.

The study shows that P. lineatus in Iskenderun Bay exhibit comparable traits to those in the Red Sea, suggesting that these regions provide suitable conditions to sustain and foster P. lineatus populations within the Mediterranean Sea.

Supplemental Information

Supplemental Information 1 Raw data.

All samples total length, weight, sex and ages.

We thank Prof. Dr Nuri BASUSTA for helping with the age determination of sagittal otoliths.

Additional Information and Declarations

Competing Interests

Author Contributions

Animal Ethics

Data Availability

Servet Doğdu is an Academic Editor for PeerJ.

Servet Ahmet Doğdu conceived and designed the experiments, performed the experiments, analyzed the data, prepared figures and/or tables, authored or reviewed drafts of the article, and approved the final draft.

Cemal Turan conceived and designed the experiments, performed the experiments, analyzed the data, prepared figures and/or tables, authored or reviewed drafts of the article, and approved the final draft.

The following information was supplied relating to ethical approvals (i.e., approving body and any reference numbers):

Iskenderun Technical University, Aquaculture Animal Experiments Local Ethics Committee has fully approved this study.

The following information was supplied regarding data availability:

The raw data is available in the Supplemental File.

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
