# Peer review of "Biological and growth parameters of Plotosus lineatus in the Mediterranean Sea"

_PeerJ, doi:10.7717/peerj.16945_

## Round 0.1 · original submission · Minor Revisions

Dear Dr. Doğdu

The reviewers have commented on your manuscript. Based on the comments and suggestions of the expert reviewers, a minor revision is needed for your article.

I would like to request that you check and correct the manuscript based on the reports. Please pay attention to the comments related to the title and self-citation.

Sincerely yours

·

Basic reporting

Dear Authors,

I have carefully reviewed your manuscript, and I appreciate the significant contribution it makes to our understanding of Plotosus lineatus dynamics in the Mediterranean Sea. The presentation of the first dataset on length-weight correlations, age-growth characteristics, and von Bertalanûy growth parameters is commendable.

Your study holds promise in providing valuable insights for the management of P. lineatus stocks, especially considering its invasive nature. However, before the manuscript can be accepted for publication, I recommend addressing a few minor issues. Ensure that the text is clear and precise. Some sentences could be refined for better readability. Elaborate further on the implications of P. lineatus' invasive nature in the context of stock management.
Once these minor issues are addressed and some other minor changes indicated, I believe your work will be well-positioned for acceptance. I look forward to seeing the revised manuscript.

Experimental design

The experimental design is succinctly presented and well-structured, contributing to the overall clarity of the study.

Validity of the findings

The work is worth to be published for its meticulous approach to ensuring the robustness and credibility of the study's outcomes. The careful use of appropriate statistical analyses adds a layer of reliability to the presented results.

Additional comments

Line 37: quote the work of Tiralongo et al., 2020, you can't be self-referential by citing only yourself. Snapshot of rare, exotic and overlooked fish species in the Italian seas: A citizen science survey. Journal of Sea Research.

Line 43: at the beginnins of the sentence, the scientific name of a species must be written in full name.
Line 58: quote Tiralongo et al., 2018. Some biological aspects of juveniles of the rough ray, Raja radula Delaroche, 1809 in Eastern Sicily (central Mediterranean Sea). Journal of Sea Research.
Line 71: scientific name in italics.

Reviewer 2 ·

Basic reporting

.

Experimental design

.

Validity of the findings

.

Additional comments

Review of the MS: Dogdu and Turan “First biological and growth parameters of Plotosus lineatus in the Mediterranean Sea”

The article reports an interesting biological data on Plotosus lineatus, an alien fish species that expands its distribution in the Mediterranean Sea. This is an additional important contribution to the knowledge this species in the northeast coast of the Mediterranean Sea.
The introduction is comprehensive and includes the required background. The figures are relevant, fig.3 is not clear (see comment below).
The research question well defined, relevant to the knowledge of the dynamic of an alien species in the Mediterranean Sea.
I recommend publishing the article after responding to the comments.

Comments

Scientific name of a species always in italics (e.g. Plotosus lineatus throughout the MS)
Line 39 – The article of Galil et al. was published in 2018 not 2017.
Line 45 - The chapter of Plotosidae in CLOFFA is written by W. R. Taylor & J. R. Gomon, not Richon. Its Distribution: “a marine species in Indian Ocean and Western Pacific as far north as Japan, sometimes entering freshwaters of East Africa and Madagascar”. The fish does not occur in Lake Malawi.
Line 72 – More details are required (mesh of inner and outer layer, size of net, day / night, month/season etc.).
Line 117 – "The majority of the population consisted of females (1:1.92). It was dominated by females". Repetition of the information
Lines 136-144 – This comparison should better be presented as a table.
Lines 148 – 153 - This comparison should better be presented as a table.
Figure 3 – The figure is not clear. The authors should replace it.

Reviewer 3 ·

Basic reporting

The title of the article is slightly misleading. As indicated within the manuscript itself, Edelist et al. (2012) investigated the length-weight relationship of this species based on specimens from the Mediterranean coasts of Israel. Therefore, it is not true that the study by Dogdu & Turan presents the "First biological and growth parameters of Plotosus lineatus in the Mediterranean Sea".

--

The article would benefit from some minor language editing. For example:

- Line 19-22: "A study was undertaken to examine the age distribution and growth characteristics of the invasive alien fish species, the striped eel catfish (Plotosus lineatus), collected from Iskenderun Bay in the eastern Mediterranean. A total of 1011 samples were obtained, with lengths ranging from 5.1 to 16.8 cm, predominantly comprising females (1:1.92)." >> "This study examined the age distribution and growth characteristics of the striped eel catfish (Plotosus lineatus), which is an invasive alien species in the eastern Mediterranean. A total of 1011 samples were collected from Iskenderun Bay (Turkey), with lengths ranging from 5.1 to 16.8 cm, predominantly comprising females (1:1.92)."

- Line 24-25: "The value of "b" was observed below "3" for females, males as and the combined sexes 2.2821, 2.2598 and 2.2737, respectively" >> "The value of the scaling exponent "b" of the length-weight relationship was less than "3" for both sexes (females: 2.28; males: 2.26; combined: 2.27)."

- "Plotosus lineatus" should be in italics [line 20, 30 in the Abtract, and also lines 42, 173, 194 elsewhere]

- Line 38-39: " To date, more than 100 Indo-Pacific fish species have been entered into the eastern Mediterranean basin via the Suez Canal" >> " To date, more than 100 Indo-Pacific fish species have entered into the eastern Mediterranean basin via the Suez Canal"

The above are all from the first page. The rest of the manuscript contains similar language issues and so the entire article requires language-editing.

--

The article includes sufficient background information to put the study into context. The relevant literature is appropriately cited and references.

--

Figures and tables are relevant to the content of the article.

Table 2 and Figure 3 need to include the units for the length measurements.

The legend for Figure 1 requires language editing for clarity and a statement of what the red crosses on the figure indicate. The text in most figures seems rather small so I recommend enlarging this to make sure it will be legible. Overall, it would be ideal to increase the resolution of the figures.

Experimental design

The study presents original research; the aim is well defined and placed within the context of gaps in knowledge.

There seem to be no major issues with the experimental design and methodology employed. The methodology is fairly standard for this type of study and could be easily replicated by other researchers.

The authors have also indicated that the study was approved by the Ethics Committee of the Iskenderun Technical University.

Validity of the findings

The study provides new and useful data on Plotosus lineatus in the Eastern Mediterranean. The raw data are all provided.

There is one minor issue with the numerical Results, which are sometimes reported with excessively long mantissas that give a false impression of accuracy. For example, the asymptotic length is given as accurate to four decimal places: 24.9934 mm. This should probably be 24.9934 cm [correction to units], but even so... the actual length measurements made during the study were only accurate to 0.1 cm, so the estimated asymptotic length should be reported as 25.0 cm or 24.99 cm at most. Likewise for "b" and "R^2" values - there is no need for more than two decimal places.

---

## Round 0.2 · accepted · Accept

Dear Dr. Doğdu

I would like to thank you and your co-authors for making the corrections and changes requested by the reviewers. I read and checked carefully your valuable article and I am happy to inform you that your article has been accepted for publication in PeerJ.

Best regards

·

Basic reporting

.

Experimental design

.

Validity of the findings

.

Additional comments

Dear Editor,

I trust this message finds you well. I wanted to inform you that the authors have diligently revised the latest version, ensuring all modifications align with the specified requirements. I suggest to accept the manuscript in its current form.

Reviewer 2 ·

Basic reporting

No further comments to the revised version. Recommend publishing it as is

Experimental design

The sampling and analysis of the data are O.K.

Validity of the findings

The analysis of the data is O.K.

Additional comments

No further comments to the revised version. Recommend publishing it as is

Reviewer 3 ·

Basic reporting

The authors have addressed the language issues that were pointed out when reviewing the oiriginal version of the manuscript.

Experimental design

The study presents original research; the aim is well defined and placed withinthe context of gaps in knowledge.

Validity of the findings

The study provides new and useful data on Plotosus lineatus in the EasternMediterranean. The raw data are all provided. The issue with excessively long mantissas has been addressed.